# Long Non-Coding RNAs as Molecular Signatures for Canine B-Cell Lymphoma Characterization

**DOI:** 10.3390/ncrna5030047

**Published:** 2019-09-19

**Authors:** Luciano Cascione, Luca Giudice, Serena Ferraresso, Laura Marconato, Diana Giannuzzi, Sara Napoli, Francesco Bertoni, Rosalba Giugno, Luca Aresu

**Affiliations:** 1Institute of Oncology Research, Universita’ della Svizzera Italiana, 6500 Bellinzona, Switzerland; sara.napoli@ior.usi.ch (S.N.); francesco.bertoni@ior.usi.ch (F.B.); 2Swiss Institute of Bioinformatics, 1000 Lausanne, Switzerland; 3Department of Computer Science, University of Verona, 37100 Verona, Italy; luca.giudice@univr.it (L.G.); rosalba.giugno@univr.it (R.G.); 4Department of Comparative Biomedicine and Food Science, University of Padova, 35100 Padova, Italy; serena.ferraresso@unipd.it (S.F.); diana.giannuzzi@phd.unipd.it (D.G.); 5Centro Oncologico Veterinario, 40037 Sasso Marconi BO, Italy; lauramarconato@yahoo.it; 6Department of Veterinary Sciences, University of Turin, 10095 Grugliasco, Italy; luca.aresu@unito.it

**Keywords:** long non-coding RNA, analysis pipeline, lymphoma, prognostic signature

## Abstract

Background: Diffuse large B-cell lymphoma (DLBCL), marginal zone lymphoma (MZL) and follicular lymphoma (FL) are the most common B-cell lymphomas (BCL) in dogs. Recent investigations have demonstrated overlaps of these histotypes with the human counterparts, including clinical presentation, biologic behavior, tumor genetics, and treatment response. The molecular mechanisms that underlie canine BCL are still unknown and new studies to improve diagnosis, therapy, and the utilization of canine species as spontaneous animal tumor models are undeniably needed. Recent work using human DLBCL transcriptomes has suggested that long non-coding RNAs (lncRNAs) play a key role in lymphoma pathogenesis and pinpointed a restricted number of lncRNAs as potential targets for further studies. Results: To expand the knowledge of non-coding molecules involved in canine BCL, we used transcriptomes obtained from a cohort of 62 dogs with newly-diagnosed multicentric DLBCL, MZL and FL that had undergone complete staging work-up and were treated with chemotherapy or chemo-immunotherapy. We developed a customized R pipeline performing a transcriptome assembly by multiple algorithms to uncover novel lncRNAs, and delineate genome-wide expression of unannotated and annotated lncRNAs. Our pipeline also included a new package for high performance system biology analysis, which detects high-scoring network biological neighborhoods to identify functional modules. Moreover, our customized pipeline quantified the expression of novel and annotated lncRNAs, allowing us to subtype DLBCLs into two main groups. The DLBCL subtypes showed statistically different survivals, indicating the potential use of lncRNAs as prognostic biomarkers in future studies. Conclusions: In this manuscript, we describe the methodology used to identify lncRNAs that differentiate B-cell lymphoma subtypes and we interpreted the biological and clinical values of the results. We inferred the potential functions of lncRNAs to obtain a comprehensive and integrative insight that highlights their impact in this neoplasm.

## 1. Background

Lymphoma is a malignant tumor that occurs in both humans and dogs as the result of a neoplastic transformation of B or T lymphocytes at different stages of development [1,2]. In veterinary medicine, B-cell lymphomas (BCL) are the most studied tumors as they represent an important comparative model for the human disease [2,3,4]. Among the different subtypes, diffuse large B cell lymphoma (DLBCL) is the most frequent in humans and dogs alike. Marginal zone lymphoma (MZL) is much more common in dogs than in humans, while the opposite is true for follicular lymphoma (FL). 

The diagnosis of lymphoma in the dog can be relatively straightforward, but the precise definition of the different BCL histotypes and the prediction of the outcome are still challenging. Similar to human lymphoma, BCLs have a heterogeneous biological and clinical behavior. In 2013, a canine DNA-microarray platform was used for the first time by Richards, and colleagues, to study DLBCL and MZL [5]. They observed a generalized NF-kB pathway activation which mirrored human activated B cell-like DLBCL (ABC-DLBCL). Six years later, a high-throughput sequencing approach (RNA-seq) successfully characterized canine DLBCL [6]; but analysis of the coding transcriptome was not able to clearly discriminate between DLBCL and MZL [7]. 

Other omics have been used to profile canine BCL and to identify potential biomarkers. A first approach profiled DLBCL with a canine-specific CpG DNA microarray [8]. DLBCLs presented a stem cell-like epigenetic signature, consistent with a high number of polycomb repressive complex targets, in stem cells affected by aberrant methylation. The analysis of epigenetic patterns and genome-wide methylation variability identified DLBCL subgroups with different outcomes, suggesting that the accumulation of aberrant epigenetic changes results in a more aggressive behavior of this tumor. Recently, high throughput sequencing technologies were applied to study methylation in canine BCL [6]. A methyl-CpG binding domain (MBD)-based approach was used to capture methylated DNA fragments, and then sequenced. The study confirmed previous data on the dysfunction of stemness genes in canine BCL, and showed that it was possible to characterize epigenetic aberrations in gene promoters at a single base level, including in introns and intergenic regions. Moreover, unsupervised hierarchical clustering of promoter methylation profiles separated dogs with DLBCL into two groups with different overall, and event free, survival. Finally, somatic mutations by whole exome sequencing were described in three pure canine breeds with different lymphoma predispositions, and genes with known involvement in human lymphoma have been described [9].

Despite increasing knowledge of canine BCL biology, the molecular mechanisms driving tumor development, and clinical outcome, are only partially understood. Moreover, the protein-coding genes that have been identified in canine BCL cannot fully explain the origin, or differences, within the different histotypes. A large portion of mammalian genomes is actively transcribed into RNA, but only about 1.5% of these transcripts are protein-coding; the remaining transcripts are transcribed into non-coding RNAs (ncRNAs). For a long time, ncRNAs received little attention, but their role in regulating complex biological processes has recently been demonstrated [10,11]. Long non-coding RNAs (lncRNAs) are a group of ncRNAs arbitrarily defined as transcripts longer than 200 bp that lack an extended protein-coding open reading frame. Even though their expression levels are generally lower compared to protein-coding genes, lncRNAs influence several aspects of cellular homeostasis, proliferation, apoptosis, and genomic stability. LncRNAs expression profiles are highly tissue-specific in both dog and human, and despite minimal overall sequence similarity, lncRNAs have evolutionarily conserved functions and regulatory effects restricted to specific cell types. This was also recently demonstrated by Le Béguec et al. [12] who identified over 900 conserved dog-human lncRNAs. In BCL, Verma et al. [13] detected lncRNAs as playing key roles in lymphomagenesis, pathogenesis and impacting on B-cell oncogenes, with a fraction being common between human and dog. 

In view of this, we explored and characterized the lncRNA landscape in canine lymphoma and sought to identify lncRNAs differentiating DLBCL, MZL and FL. 

## 2. Methods

The two previously published pipelines for lncRNAs detection consisted of aligning and assembling transcripts, creating a unique transcriptome, comparing the data with the annotated genome, and finally evaluating the coding potential of each novel transcript in order to separate coding and non-coding transcripts. Verma et al. [13] used CPAT [14], which is generally suggested for well annotated organisms; whereas Wucher et al. [15] proposed a new method to identify lncRNAs when genomes are not well annotated, or where a list of annotated lncRNAs is not available. 

Here, we describe a modified pipeline for the detection and analysis of novel and annotated lncRNAs in canine BCL (see Figure 1). The dataset included dogs affected by DLBCL (n = 50), FL (n = 7), and MZL (n = 5) that had undergone complete staging work-up and were treated with chemotherapy or chemo-immunotherapy. All dogs underwent lymphadenectomy to confirm lymphoma histotype by routine histology and immunohistochemistry, and a portion of the neoplastic lymph node was preserved in RNA-later for research purposes. Normal canine B-cells were obtained from the lymph nodes of 13 clinically normal dogs. The B-cell compartment was isolated by laser capture microdissection, and then frozen preserved. Approval for this study was granted by MIUR Ethical Board (Number RBSI14EDX9).

Library preparation and sequencing approach have been described elsewhere [6,16]. Reads were deposited in GenBank (accession numbers SRP137798 and SRP140599).

Illumina reads were first preprocessed with FastaQC software in order to assess the quality of the sequences. Reads were aligned with STAR [17] using the canine reference genome (CanFam3.1.87), assembled in de novo mode to find novel and annotated transcripts, and then merged by StringTie [18]. StringTie was preferred to CuffLink because of its higher speed and comparable accuracy [19]. Cuffcompare, a tool of the CuffLinks suite, was used to compare the consensus transcriptome outputted by StringTie to CanFam3.1.87. By using parameters -M and -R, the process was accelerated since both monoexonic transcripts and transcripts present only in the reference annotation were not considered. The comparison step detected all the protein coding genes, annotated lncRNAs, and other transcripts. The annotated transcripts were then filtered out maintaining only the ones flagged as unknown and considered as potential novel lncRNAs (see Figure 2). Transcripts in chromosomes 1-38 or chromosome X, containing at least 2 exons, and with an exon length greater than 200 nucleotides were maintained. The resulting transcriptome was converted into a FASTA file and given as input to FEELnc_codpot [15], which estimated the coding potential of transcripts. As input, FEELnc required the two FASTA files containing the protein-coding and the lncRNAs sequences. It produced a list of all the transcripts associated to their corresponding coding potential and the cutoff that separated coding from non-coding transcripts. Further, transcripts were validated by using the command line version of BLAST and novel and annotated transcripts were quantified by using htseq-count in single-stranded mode by HTSeq software package [20]. Only reads that were uniquely aligned were retained. Differential expression analysis was performed in R, using the limma [21] and edgeR [22] packages. Only novel transcripts with cpm ≥ 10, and annotated ones with cpm ≥ 1 in at least 3 samples, were considered [23]. Deregulated expression of transcripts was considered when FDR <0.05 (*p*-value adjusted for multiple comparisons). Tumor subtyping was performed by using R-ISDBSCAN clustering algorithm [24], which is an extension of the density-based clustering algorithm DBSCAN [25]. Density-based algorithms compared to partition-clustering methods, such as k-means, were able to automatically detect the number of subtypes and noise elements (outliers) in the dataset. R-ISDBSCAN outperformed DBSCAN in the accuracy of group identification, handling local point density changes more appropriately.

Overall survival (OS) was defined as the time from diagnosis to death, and event-free survival (EFS) was defined as the time from starting therapy to the date of any diagnosed relapse. Survival was estimated with the Kaplan-Meier method [26] and compared by log-rank test [27]. P values less than 0.05 were considered statistically significant. The *p*-values for multiple comparisons were adjusted using the Benjamini–Hochberg correction. The performance of the lncRNAs model and protein-coding genes signature for OS and EFS was compared by a measure of global fit (AIC) and a measure of discrimination (CPE) along with its 95% CI [28,29,30]. Low AIC values indicated a better fit, and high CPE values discriminated better.

For functional analysis of lncRNAs, a guilty-by-association approach was used where all the protein-coding genes close to lncRNAs were retrieved using FEELnc_classifier [15]. All upstream and downstream transcripts within 10 kilobases were considered because of the long-range activity of cis-molecules [31]. Enrichment analysis of the contrasts of interest was performed by DAVID [32], giving the set of lncRNAs associated protein-coding genes as input. Functional annotation was performed using the Gene Set Enrichment Analysis (GSEA) tool for overlap analysis and the hallmark gene set, the c2.cp gene set of the Molecular Signatures Database (MSigDB) 5.2 [33], and custom gene sets of the IOR Institute [34]. Moreover, the protein-protein interaction network was built using the same input and cuRnet, an R package that provides GPU solutions of algorithms to analyze biological networks, was run to identify high-connected subnetworks potentially belonging to the key pathways that are altered in a studied condition [35]. cuRnet generally solves this issue running a parallel strong connected component algorithm and interactively visualizes the resulting subnetworks.

## 3. Results

In order to get an exhaustive catalog of canine lncRNAs we first performed a systematic transcript discovery using a dataset comprising lymphomas (n = 62) and controls (n = 13). A total of 4478 putative unknown lncRNAs were obtained and found expressed. After filtering in length and coding potentiality, the number decreased to 1629 lncRNAs, but only 781 were annotated. 

Using two previously obtained human and canine BCL datasets (phs000235.v6.p1, SRP021509 and SRA059558), a total of 3.5% and 67% of lncRNAs were found in common, respectively (Figure 2A). Next, we assessed the differential expression (DE) of both novel, and already annotated lncRNAs in tumors, and controls. The two groups clustered separately when considering both novel lncRNAs and annotated lncRNAs (Figure 2B). This result demonstrated a good performance of the method used, and the biological utility of lncRNAs in differentiating the two classes. Overall, lymphomas were characterized by a higher number of down-regulated transcripts (Appendix A). When we performed DE analysis separating tumors by histotypes, the number of lncRNAs differentially expressed in DLBCLs was markedly higher compared with FL and MZL.

A total of 144 novel lncRNAs were found differentially expressed in DLBCL and MZL. Conversely, only one novel lncRNA transcript was differentially expressed when DLBCL and FL were compared. A moderate number of differentially expressed lncRNAs was also found when comparing MZL with FL, but the number of cases in the two datasets was too low to obtain further information. Figure 3 shows the number of differentially expressed lncRNAs, and a list of all the transcripts is provided in Appendix A.

We investigated the putative function of differentially expressed lncRNAs by associating protein-coding genes to lncRNAs within 10 kilobases. We then applied three strategies to identify significantly correlated pathways (Appendix A). 

The analysis showed significant enrichment of GO and KEGG terms among lymphoma subtypes. Modulated lncRNAs in DLBCL vs Control, and MZL vs Control were primarily involved in the regulation of cell proliferation, chromatin silencing, cell death and transcriptional misregulation (Figure 4).

Next, considering the high clinical and biological heterogeneity of canine DLBCL, we applied a density-based algorithm to identify lncRNAs signatures affecting the clinical behavior of this histotype. Three distinct subgroups, comprising 44 DLBCLs were obtained and named DLBCL1, DLBCL2, and DLBCL3. Six tumors were labelled as noise by the algorithm and were therefore excluded. The three groups reflected different clinical behaviors with dogs in the DLBCL1 group, being characterized by significantly shorter OS and EFS compared to dogs in DLBCL2 and DLBCL3 groups (Figure 5). Also, differential expression analysis between DLBCL1 and DLBCL2 showed a higher number of differentially expressed lncRNAs than other comparisons (Figure 6). These results underline the utility of lncRNA profiles in defining clinically diverse groups.

Previous studies with DNA microarray and next generation sequencing had not identified differences between MZL and DLBCL [4,5,6,7,16]. Therefore, we compared MZL against each individual DLBCL group hypothesizing differences and similarities among groups. Differential expression analysis determined a complete overlap of MZL with DLBCL1 and DLBCL3, but not with DLBCL2 (Figure 7).

Previously, when we only considered the expression of protein-coding genes in the same dataset of DLBCLs, we obtained two groups named EHA1 and EHA2 [6]. The EHA2 group was characterized by a poorer outcome and showed an enrichment of T-cell-mediated immune response signatures, compared to EHA1 [6]. The lncRNAs-driven clusters identified here, partially overlapped with EHA1 and EHA2. Seventy-five percent of the dogs in the DLBCL1 group were included in the EHA2 group, while dogs in DLBCL2, DLBCL3, Noise, and the remaining 25% of dogs in the DLBCL1 group belonged to the EHA1 group (Appendix A). Dogs in DLBCL1 that were split between EHA1 and EHA2 did not show significant changes for EFS and OS. DLBCL1 represented a homogenous diagnostic group compared with non-DLBCL1 (see Figure 5), and this was also confirmed by GSEA analysis (Appendix A) comparing DLBCL1 in EHA1 versus DLBCL1 in EHA2, DLBCL1 in EHA1 versus DLBCL2, and DLBCL3 versus DLBCL1 in EHA2. The Cox model built using lncRNAs showed a CPE equal to 0.61 (95% CI 0.58–0.64), suggesting that our non-coding predictor had better discriminatory power than the protein-coding one (0.56, 95% CI 0.52–0.60). Similar conclusions were reached using the global model fit criterion (AIC), with the prognostic classifier based on lncRNAs, achieving a better global model fit compared to protein-coding (258.26 vs. 255.35).

## 4. Discussion

In human medicine lncRNAs are considered fundamental in many biological processes and are often dysregulated in cancers. Little is known about lncRNAs in dogs and the challenge when studying lncRNAs is their relatively low abundance and reduced conservation across species. Although previous studies have demonstrated the involvement of lncRNAs in B-cell lymphoma, comprehensive characterization of the transcriptome, prognostic role, and functional contribution of lncRNAs in distinct B-cell subtypes are lacking. Our analysis adds significant and novel insights by providing the most comprehensive dataset so far for canine DLBCL, MZL and FL. We identified differentially expressed subtype-specific lncRNAs and investigated their putative function by associating protein-coding genes and pathways. GO and KEGG pathway analysis revealed that differentially expressed lncRNAs served as regulators of cell proliferation, transcription, and cell cycle (Appendix A). The dysregulation of pathways related to B-cells has been consistently observed both in human and canine lymphomas and here we show that when considering expressed lncRNAs the similarity is maintained between the two species. These results reinforce the dog as a valid large animal model when studying B-cell lymphoma. Moreover, a lncRNAs-based clustering of DLBCL cases largely overlapped with what we recently achieved using protein coding genes [6], albeit it with a better identification of subgroups with different outcomes.

One of the most interesting results of this work was obtained when comparing MZL to DLBCL subgroups. Recent literature has defined MZL as a continuum of DLBCL mostly because the transcriptomic profiles are very similar [5]. Our work confirms this aspect, but also identified differences. In fact, a subgroup of DLBCL, named as DLBCL2 cluster, did present a moderate number of lncRNAs differentially expressed compared to MZL. This observation suggests that only a fraction of DLBCL might be directly related to MZL, possibly resembling what has recently been reported in human DLBCL with the identification of DLBCL subtypes bearing genetic lesions, such as NOTCH activation, that are typically present in MZLs [36,37].

## 5. Conclusion

In conclusion, our study provides an in-depth analysis of the lncRNAs transcriptome in canine B-cell lymphoma subtypes. Our analysis underlines the biological and prognostic role of lncRNAs in this disease. LncRNAs, quantified by our pipeline, clearly separate normal from pathological samples and uncover previously unidentified differences between DLBCL and MZL. We also found clusters with prognostic value within the DLBCL histotype: lncRNAs profiling robustly identified, significantly different, subgroups and identified the DLBCL1 subgroup as having a higher mortality rate than DLBCL2 and DLBCL3. Thus, our results provide a basis for further studies to characterize the lncRNA profiles of dogs with a poor prognosis, with the aim of identifying possible predictive biomarkers that can be utilized at the time of diagnosis. Finally, the findings of this work support the broad translational application of canine hematological disorders as comparative models for human B-cell lymphoma.

## Figures and Tables

**Figure 1 ncrna-05-00047-f001:**
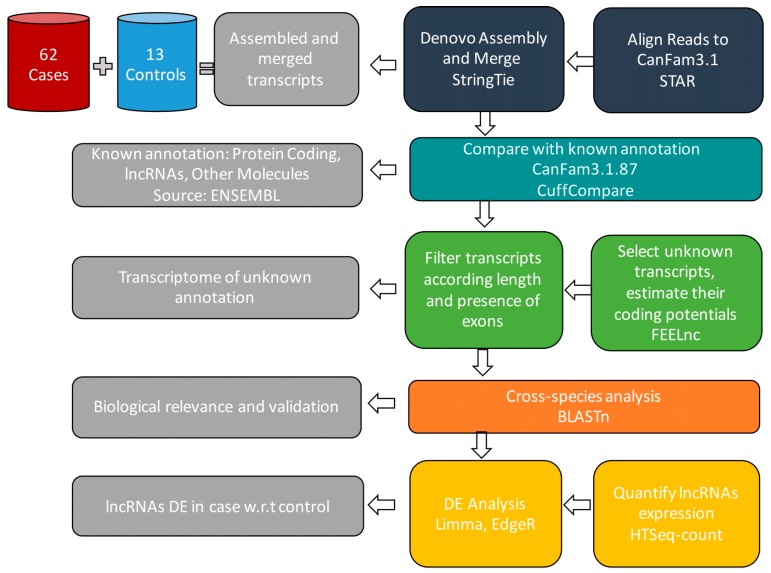
Diagram of the RNA-sequence analysis workflow. Illumina reads are aligned with STAR using as reference genome CanFam3.1.87, assembled to discover novel and annotated transcripts, and then merged by StringTie. The output consensus transcriptome is compared to CanFam3.1.87 by Cuffcompare to detect protein coding genes, lncRNAs, and other molecules. The known transcripts are filtered out to keep novel lncRNAs with enough coding potential computed using FEELnc_codpot. Finally, transcripts are validated using BLAST and both the novel and the annotated ones are quantified using HTSeq-count. The resulting lncRNAs, with their related counts, are analysed by Limma and EdgeR to identify differentially expressed ones.

**Figure 2 ncrna-05-00047-f002:**
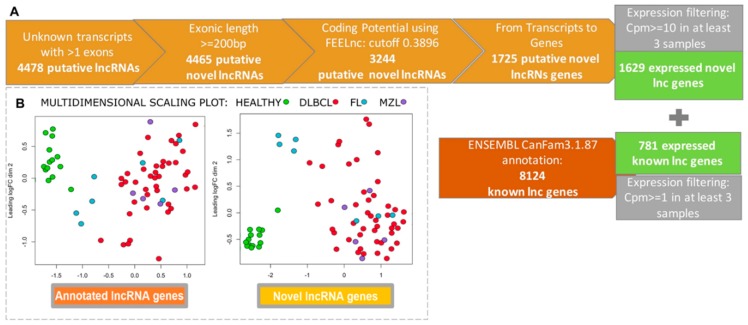
(**A**) Filtering steps applied to the initially discovered novel transcripts to identify bona fide novel lncRNAs, based on exon count, transcript length, and non-coding potential using FEELnc. Post these, expression level filtering (cpm ≥ 10 in 3 samples or more) was performed. (**B**) Multidimensional scaling (MDS) plot of RNA-Seq data for: annotated lncRNA genes (left) and novel lncRNA genes (right). Each dot represents a sample, which is colored by its histology.

**Figure 3 ncrna-05-00047-f003:**
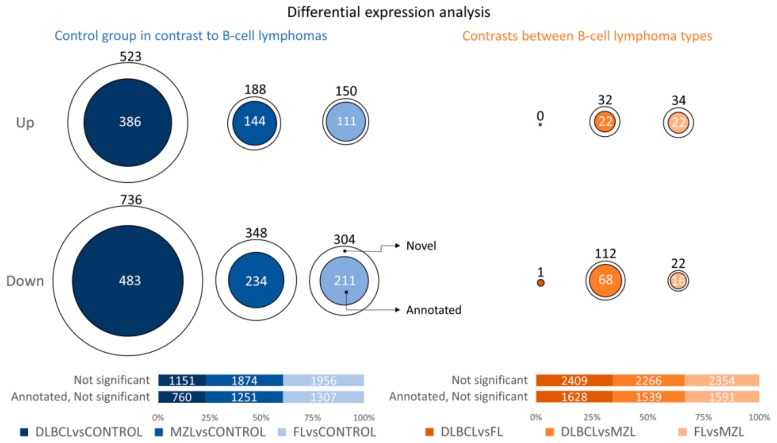
The number of differentially expressed lncRNAs (up: upregulated, down: downregulated) and the percentage of not differentially expressed novel and annotated lncRNAs between Control versus B-Cell lymphoma histotypes (blue part of the plot: DLBCL vs. Control, MZL vs. Control, FL vs. Control) and pairwise contrast among B-cell histotypes (orange part of the plot: DLBCL vs. FL, DLBCL vs. MZL and FL vs. MZL).

**Figure 4 ncrna-05-00047-f004:**
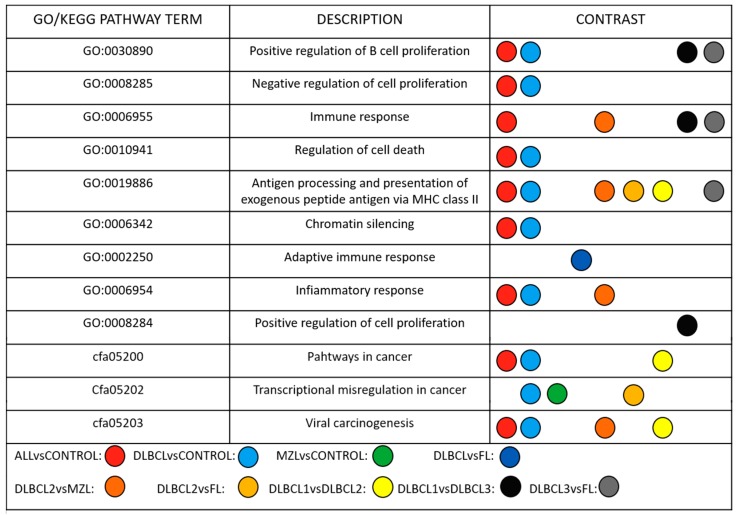
Potential functions and putative pathways of deregulated lncRNAs in the comparison of interest. Each colored circle represents a comparison and, if present in one row, shows the significant enrichment of the modulated lncRNAs for that comparison for the specific GO term or KEGG pathway.

**Figure 5 ncrna-05-00047-f005:**
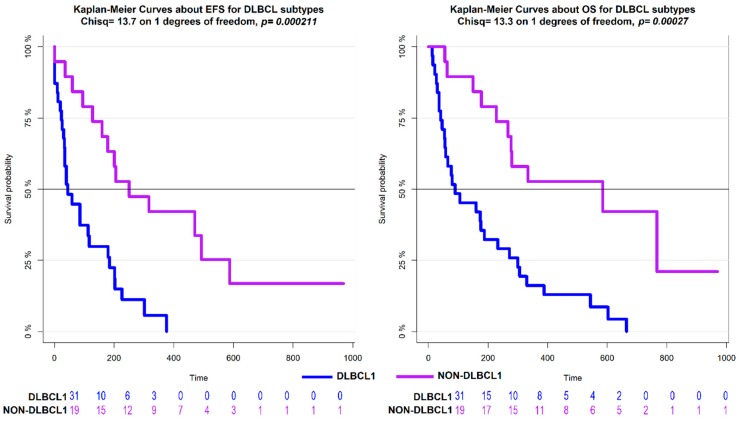
Kaplan–Meier estimates of Event Free Survival (EFS, left part) and Overall Survival (OS, right part) according to DLBCL clusters based on lncRNA expression profiles.

**Figure 6 ncrna-05-00047-f006:**
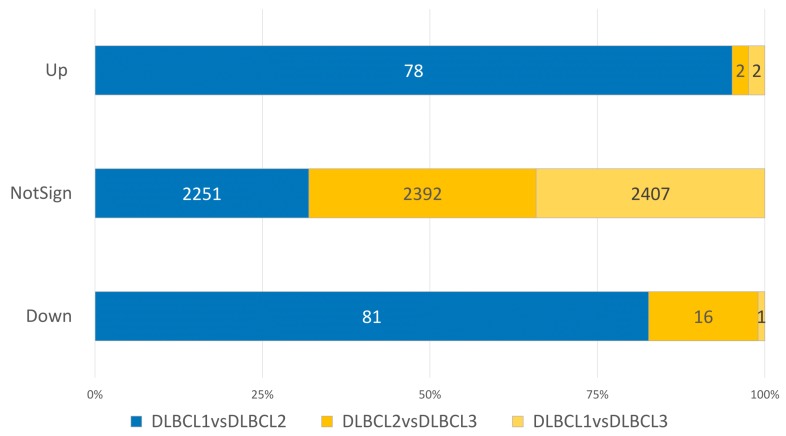
Percentage of differentially expressed lncRNAs in pairwise comparison among DLBCL clusters (DLBCL1 vs. DLBCL2, DLBCL2 vs. DLBCL2, and DLBCL1 vs. DLBCL2). The three stack bar graph represents Upregulated (Up), not statistically significant modulated (NotSign), and Downregulated (Down) molecules. Each colored proportion of a bar is equal to the percentage of the number of lncRNAs respect of the total amount for deregulated molecules.

**Figure 7 ncrna-05-00047-f007:**
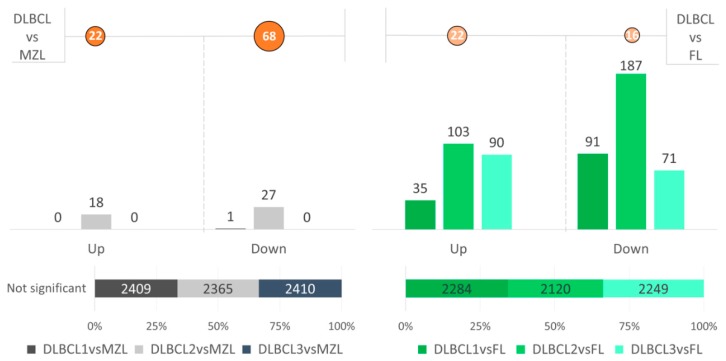
The number of differentially expressed lncRNAs (up: upregulated, down: downregulated) and the percentage of not differentially expressed lncRNAs between DLBCL vs. MZL, and pairwise comparison of DLBCL subtypes (DLBCL1, DLBCL2 and DLBCL3) versus MZL. The orange circles refer to the results of the DE analysis among the overall groups (e.g., DLBCL versus MZL has 22 upregulated and 68 downregulated lncRNAs).

## Data Availability

Sequencing data is deposited on the SRA database under the accession number SRP137798.

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
