# Peer review of "Long Non-Coding RNAs as Molecular Signatures for Canine B-Cell Lymphoma Characterization"

_ncrna, 2019, doi:10.3390/ncrna5030047_

Round 1
Reviewer 1 Report
In this report Cascione et al. perform an exhaustive analysis of lncRNAs in canine B cell lymphoma. They develop a pipeline a comprehensive pipeline to characterize the lncRNAs that allow the identification of 2 subtypes of DLBCL. The authors very nicely infer the putative function of the differentially expressed lncRNAs. In my opinion, this is a thorough work that might add some knowledge to the field. However, some minor improvements should be made. In particular, the legends of the figures should be improved. In most of the figures they do not describe what the figure shows.
-Figure 2B. Maybe easier to just show coloured dots instead of letters (DLBCL, FL…) It gets very messy. The legends do not explain what the graphs contain
-Figure 3. Legends need to be improved. What is blue and what is orange?
-The data related to Figure 7 and 8 should be included in results and then discussed in the discussion part.
-Figure 8 should go in supplementary material.
Author Response
Response to Reviewer 1 Comments
Point 1: The legends of the figures should be improved. In most of the figures they do not describe what the figure shows.
Response 1: We modified the legends of the figures.
Point 2: Figure 2B. Maybe easier to just show coloured dots instead of letters (DLBCL, FL…) It gets very messy. The legends do not explain what the graphs contain
Response 2: We modified the figure, now the plot shows dots instead of letters and we improve the legend.
Point 3: Figure 3. Legends need to be improved. What is blue and what is orange?
Response 3: Legends have been modified clarifying what the plot contains.
Point 4: The data related to Figure 7 and 8 should be included in results and then discussed in the discussion part.
Response4: Data relative to Figure 7 and 8 have been included in the results section and discussed in the discussion part.
Point 5: Figure 8 should go in supplementary material.
Response 5: We removed the Figure 8 from the text.
Reviewer 2 Report
The article is a really interesting one, and brings new and important information about the molecular biology of canine B lymphoma. Methods are adequate, and some minor revision are important.
Introduction:
This was also demonstrated recently by Le Béguec, C. et al
In BCL, Verma A et al. [13]
Methods:
whereas Wucher V. et al. [15] proposed
Please do not use the first letter of the name of the authors
Methods: please, use past tense. Be more specific from normal B cells were collected. From blood? Lymph node?
How RNA was extracted from samples? Were the lymphoma samples frozen or FFPE?
Discussion:
It should be improved. It looks like results and does not bring discussion about the pathways and the role of lncRNAs in the prognosis and pathogenesis of canine B lymphoma. Discuss more about the translational aspect and comparative oncology model of this work .
Conclusion: should be more straightforward, focusing in the aims of the present work. For example: what were the prognostic cluster found?
Author Response
Response to Reviewer 2 Comments
Point 1: Please do not use the first letter of the name of the authors
Response 1: We modified the text according to reviewer’s suggestion.
Point 2: Methods: please, use past tense.
Response 2: We modified the text using the past tense in the Methods section.
Point 3: Be more specific from normal B cells were collected. From blood? Lymph node? How RNA was extracted from samples? Were the lymphoma samples frozen or FFPE?
Response 3: We specify in the text how normal B cells were collected, how the RNA was extracted and if lymphoma samples are frozen or FFPE. The samples were used in our previous publication [Aresu et al., Heamatologica 2018] and uploaded into SRA database (https://trace.ncbi.nlm.nih.gov/Traces/sra/?study=SRP137798). More details about samples library preparation could be found in the on line supplementary material of the manuscript published in Heamatologica and on the SRA study web site (https://trace.ncbi.nlm.nih.gov/Traces/sra/?study=SRP137798)
Point 4: Discussion should be improved. It looks like results and does not bring discussion about the pathways and the role of lncRNAs in the prognosis and pathogenesis of canine B lymphoma. Discuss more about the translational aspect and comparative oncology model of this work.
Response 4: We modified the Discussion and Results section moving some part from discussion to Results. We added and modified some paragraphs in both sections.
Point 5: Conclusion: should be more straightforward, focusing in the aims of the present work. For example: what were the prognostic cluster found?
Response 5: We modified the Conclusion focusing on the aim of the word, and got it a little bit more straightforward.